# The Application of Quaternions to Strap-Down MEMS Sensor Data

**DOI:** 10.3390/s21227658

**Published:** 2021-11-18

**Authors:** Peter M. Dickson, Philip J. Rae

**Affiliations:** Los Alamos National Laboratory, Los Alamos, NM 87545, USA; prae@lanl.gov

**Keywords:** quaternion, MEMS, 6-axis, blast loading, animation

## Abstract

We describe the mathematical transformations required to convert the data recorded using typical 6-axis microelectromechanical systems (MEMS) sensor packages (3-axis rate gyroscopes and 3-axis accelerometers) when attached to an object undergoing a short duration loading event, such as blast loading, where inertial data alone are sufficient to track the object motion. By using the quaternion description, the complex object rotations and displacements that typically occur are translated into the more convenient earth frame of reference. An illustrative example is presented where a large and heavy object was thrown by the action of a very strong air blast in a complex manner. The data conversion process yielded an accurate animation of the object’s subsequent motion.

## 1. Introduction

### 1.1. Strap-Down Microelectromechanical Sensors

Strap-down microelectromechanical systems (MEMS) are typically 6-axis microscopic sensor packages that, when attached to rigid objects, are able to measure acceleration and angular velocity in the object frame of reference (FOR) [1]. Reduction in the resulting time-resolved data is complicated by the moving frame of reference, and almost always requires transformation into a suitable fixed FOR. While there are various methods available to do that, quaternion representations are the most efficient.

For applications such as flight control or biomechanical tracking, in which the motion is followed for durations of minutes or more, the cumulative errors arising from integration of drift and bias errors, especially in the double integration of the accelerometer data, require some form of correction from absolute, rather than differential, data sources. Absolute rotation data around the vertical axis are often supplied by magnetometers, with global navigation satellite system (GNSS) data used to correct absolute position. These additional data are typically applied via a Kalman filter [2] or similar algorithm.

This application, however, involves large-magnitude, short-duration accelerations and rotations driven by blast and/or impact events. This method has been used for tracking munition launch dynamics [3], but here we apply it to even shorter impulse-driven events.

The use of a 6-axis MEMS system safely contained within the test article when undertaking blast loading is a considerable advantage. The shock wave, explosive reaction products, or dust kicked up by the blast are major problems for the only comparable technique for resolving full motion in three-dimensions: orthogonal high-speed image analysis. Optical techniques in these conditions suffer from potential camera damage, severe image obstruction from smoke and dust, and positioning suitable long duration high-intensity ancillary lighting. Additionally, motion resolution from image analysis is a complex tradeoff between the times of frame exposure and interframe, as well as the dimensions of the field of view. Finally, the double differentiation required of measured object displacements to yield initial accelerations results in large uncertainties in the measured values during high magnitude and short duration events, such as blast loading.

Although all the mathematical derivations described in what follows are available in separate pieces in prior reports and books, we have not found a single approachable and rigorous raw data input to the earth’s FOR publication and so we believe that such a summary would prove useful to others. Additionally, available descriptions for longer duration events often already include the necessary, but complex, correction terms for the Kalman filter.

### 1.2. Introduction to Quaternions

Euler’s rotation theorem states that, in three-dimensional space, any sequence of rotations of a rigid body or coordinate system about a fixed point can be achieved by a single rotation through an angle, θ, about an axis (the *Euler axis*) that runs through that fixed point.

Quaternions (more specifically *unit quaternions*) provide an easy way to describe this axis–angle representation of a rotation [4]. If the quaternion represents the orientation of an object in a coordinate system, then it is often referred to as an *orientation quaternion*, whereas if it is used to rotate an object from its existing orientation, then it is called a *rotation quaternion*. While these are functionally identical, orientation quaternions are generally applied to vectors to transform their frame of reference, while rotation quaternions are applied to orientation quaternions to get new orientation quaternions.

If the Euler axis is represented by a unit vector, **u**, the rotation can be represented as a combination of the scalar θ and vector **u**. The quaternion, **q**, appears in the unusual form of the sum of a scalar, *s*, and vector, **v**
q=s+v
where the scalar is the real part and the vector is the imaginary part, and it is usually written as
(1)$$q=q_{0}+q_{1}i+q_{2}j+q_{3}k.$$

In this notation, **i**, **j**, and **k** are equivalent to the Cartesian unit vectors but are the basic quaternion units that have the unusual, non-commutative, multiplicative properties, known as the Hamilton products
$$i^{2}=j^{2}=k^{2}=ijk=-1,$$


ij=k, ji=−k, jk=i, kj=−i, kj=j, ik=−j


With these rules applied the product of two quaternions, **p** and **q**, may be written as
(2)$$pq=\left(p_{0}+p_{1}i+p_{2}j+p_{3}k\right)\left(q_{0}+q_{1}i+q_{2}j+q_{3}k\right)\;=\left(p_{0}q_{0}-p_{1}q_{1}-p_{2}q_{2}-p_{3}q_{3}\right)+\left(p_{0}q_{1}+p_{1}q_{0}+p_{2}q_{3}-p_{3}q_{2}\right)i+\;\;\left(p_{0}q_{2}-p_{1}q_{3}+p_{2}q_{0}+p_{3}q_{1}\right)j+\left(p_{0}q_{3}+p_{1}q_{2}-p_{2}q_{1}+p_{3}p_{0}\right)\mathrm{k.}$$

It is straightforward, by inspection, to show that this is equivalent to
(3)$$pq=p_{0}q_{0}-p\cdot q+p_{0}q+q_{0}p+p\times q$$
where
$$p\cdot q=p_{1}q_{1}+p_{2}q_{2}+p_{3}q_{3}\;\mathrm{and}\;p\times q=\left|\begin{array}{ccc} i & j & k\\ p_{1} & p_{2} & p_{3}\\ q_{1} & q_{2} & q_{3} \end{array}\right|.$$

This product of two unit quaternions is also a unit quaternion and, alternatively, can be conveniently represented in matrix notation as
(4)$$pq=r=\;[\begin{array}{c} r_{0}\\ r_{1}\\ r_{2}\\ r_{3} \end{array}]=\;[\begin{array}{cccc} p_{0} & -p_{1} & -p_{2} & -p_{3}\\ p_{1} & \;p_{0} & -p_{3} & p_{2}\\ p_{2} & \;p_{3} & \;p_{0} & -p_{1}\\ p_{3} & -p_{2} & \;p_{1} & p_{0} \end{array}]\;[\begin{array}{c} q_{0}\\ q_{1}\\ q_{2}\\ q_{3} \end{array}].$$

The product quaternion, r, is the quaternion representing the rotation of **q** followed by **p**.

Since **q** is a unit quaternion its inverse is the conjugate quaternion, **q***, and is found simply by changing the sign of the imaginary part
$$q^{*}=q_{0}-q_{1}i-q_{2}j-q_{3}k.$$

In the notation introduced above, the unit quaternion, **q**, representing a rotation of *θ* about the axis, **u**, is constructed as
(5)$$q\;=e^{\frac{\theta }{2}\left(u_{x}i+u_{y}j+u_{z}k\right)}=\cos\frac{\theta }{2}+\sin\frac{\theta }{2}\left(u_{x}i+u_{y}j+u_{z}k\right)/\|u\|.$$

Rotation of a vector in Cartesian space, $v=v_{x}i+v_{y}j+v_{z}k$, to a new vector, $v^{\prime }$ is achieved by
(6)$$v^{\prime }=qvq^{*}$$
where the vector, **v**, is treated as a quaternion with real part $v_{0}$= 0, giving
(7)$$v^{\prime }=\left(q_{0}+q_{1}i+q_{2}j+q_{3}k\right)\left(0+v_{x}i+v_{y}j+v_{z}k\right)\left(q_{0}-q_{1}i-q_{2}j-q_{3}k\right).\;$$

This can be more usefully expressed as a 3 × 3 rotation matrix, R, where
(8)$$v^{\prime }=Rv=\;[\begin{array}{ccc} 1-2\left(q_{2}^{2}+q_{3}^{2}\right) & 2\left(q_{1}q_{2}-q_{3}q_{0}\right) & 2\left(q_{1}q_{3}+q_{2}q_{0}\right)\\ 2\left(q_{1}q_{2}+q_{3}q_{0}\right) & 1-2\left(q_{1}^{2}+q_{3}^{2}\right) & 2\left(q_{2}q_{3}-q_{1}q_{0}\right)\\ 2\left(q_{1}q_{3}-q_{2}q_{0}\right) & 2\left(q_{2}q_{3}+q_{1}q_{0}\right) & 1-2\left(q_{1}^{2}+q_{2}^{2}\right) \end{array}]\;[\begin{array}{c} v_{x}\\ v_{y}\\ v_{z} \end{array}],$$
or in the orthogonal form
(9)$$v^{\prime }=Rv=\;[\begin{array}{ccc} q_{0}^{2}+q_{1}^{2}-q_{2}^{2}-q_{3}^{2} & 2\left(q_{1}q_{2}-q_{3}q_{0}\right) & 2\left(q_{1}q_{3}+q_{2}q_{0}\right)\\ 2\left(q_{1}q_{2}+q_{3}q_{0}\right) & q_{0}^{2}-q_{1}^{2}+q_{2}^{2}-q_{3}^{2} & 2\left(q_{2}q_{3}-q_{1}q_{0}\right)\\ 2\left(q_{1}q_{3}-q_{2}q_{0}\right) & 2\left(q_{2}q_{3}+q_{1}q_{0}\right) & q_{0}^{2}-q_{1}^{2}-q_{2}^{2}+q_{3}^{2} \end{array}]\;[\begin{array}{c} v_{x}\\ v_{y}\\ v_{z} \end{array}].$$

It follows also that if **q** is the orientation quaternion for an object in Cartesian space, then Equations (6)–(9) also represent the transformation of vectors from the object’s FOR into Cartesian coordinates.

### 1.3. Euler Angles

While quaternions provide a very concise method to describe and execute rotations, they do not represent an intuitive way to visualize them. The more conventional Euler angles are still useful for that purpose, and so the relationship between quaternions and Euler angles is important. Euler angles are based on the use of three rotations, each about a principal axis, to reach a target orientation, but the choices of axes, extrinsic or intrinsic rotations (Extrinsic—rotation about a coordinate system axis; intrinsic—rotation about an object axis), and the order of rotations, result in numerous possible definitions.

A commonly used set of angles are the Tait–Bryan angles that were developed for aerospace applications, and in the earth’s FOR are referred to as heading (*Ψ*), elevation (*θ*), and bank (φ). Colloquially, the terms yaw, pitch, and roll are also used, although those are really incremental changes in attitude around the axes, as shown in Figure 1.

In the fixed frame of reference comprising *x*, *y*, and *z* axes, it is conventional that the *z*-axis points down. In the earth’s FOR, *x* is north and *y* is east. The rotations are generally considered as executed intrinsically on an object with principal axes *X*, *Y*, and *Z* that start aligned with the FOR axes, with *X* representing forwards, *Y* to the right, and *Z* down.

Yaw is first, clockwise around the FOR *z*-axis (which is coincident with the initial object *Z*-axis), followed by pitch rotation clockwise about the new object *Y*-axis, followed by roll clockwise at the new object’s *X*-axis. However, note that these are exactly equivalent to the rotations executed extrinsically about the fixed axes of the FOR but in the opposite order, i.e., roll about *x* followed by pitch about *y* followed by yaw about *z*.

Heading can be expressed in the range 0°–360° or −180°–180°. In this application, the latter is used.

The Tait–Bryan angles are computed from the orientation quaternion in Equation (1) as follows
(10)$$\psi =\arctan2\left(2q_{1}q_{2}+2q_{3}q_{0},1-2\left(q_{2}^{2}+q_{3}^{2}\right)\right)$$
(11)$$\theta =\arcsin\left(2q_{2}q_{0}-2q_{1}q_{3}\right)$$
(12)$$\phi =\arctan2\left(2q_{0}q_{1}+2q_{2}q_{3},1-2\left(q_{1}^{2}+q_{2}^{2}\right)\right)$$
where the arctan2 function is used, as usual, to avoid incorrect solutions at *Ψ* ≥ 90° and *φ* ≥ 90°.

Conversely, the rotation quaternion corresponding to specified Tait–Bryan angles is found by considering the individual rotations that they represent in the order that they occur. Using the extrinsic rotation form, from Equation (5)
$$q_{x}=\cos\frac{\phi }{2}+i\sin\frac{\phi }{2}$$
$$q_{y}=\cos\frac{\theta }{2}+j\sin\frac{\theta }{2}$$
$$q_{z}=\cos\frac{\psi }{2}+k\sin\frac{\psi }{2}$$
and so
$$q=q_{z}q_{y}q_{x}=q_{0}+q_{1}i+q_{2}j+q_{3}k$$
giving
(13)$$q_{0}=\cos\psi \cos\theta \cos\phi +\sin\psi \sin\theta \sin\phi$$
(14)$$q_{1}=\cos\psi \cos\theta \sin\phi -\sin\psi \sin\theta \cos\phi$$
(15)$$q_{2}=\cos\psi \sin\theta \cos\phi +\sin\psi \cos\theta \sin\phi$$
(16)$$q_{3}=\sin\psi \cos\theta \cos\phi -\cos\psi \sin\theta \sin\phi$$

## 2. Application to Object for Inertial Data

The preceding relationships provide the necessary framework to transform strap-down sensor inertial data into the earth or laboratory’s FOR and integrate with respect to time to get heading, attitude, velocity, and position as functions of time.

These sensors, which may be rigidly or semi-rigidly mounted to approximately rigid bodies, measure and record acceleration and angular velocity, i.e., a(t) and ω(t), respectively, on their three principal axes as $a_{X},\;a_{Y},\;a_{Z},\;\omega_{X},\;\omega_{Y},\;\omega_{Z}$ at a sample rate *s* and time interval ∆*t* = 1/s.

The basic strategy is to start with the orientation quaternion that describes the initial orientation of the principal axes of the object (generally some permutation of the sensor axes *x*, *y*, and *z*) in the fixed frame of reference in use (*X*, *Y*, *Z*); iteratively transform the sensor acceleration and angular velocity data into the fixed frame of reference to get $a_{x},\;a_{y},\;a_{z},\;\omega_{x},\;\omega_{y},\;\omega_{z}$; and then use the resulting angular velocities to update the orientation quaternion. Once this was completed, the accelerations are integrated with respect to time, once to get velocity and twice to get position.

There are a few additional details to attend to. These sensors are typically supplied with scaled outputs and nominally zero bias but often, especially in the case of high-dynamic-range sensors, the data must be zeroed before the start of motion. If the data start with the object stationary, which is the preferred initial condition, then angular velocities can be compensated to zero, but the accelerometers will measure a constant acceleration of one g upwards, the components of which in the sensor’s FOR must be subtracted before zeroing. This requires knowledge of the object orientation, typically in the form of the Tait–Bryan angles, before motion starts.

If initial orientation is not known, then it can be estimated from the initial accelerometer data if the accelerometers are calibrated in advance, assuming that the biases have not drifted. A necessary, but not necessarily sufficient, condition in that case is
$$\sqrt[{2}]{a_{X}^{2}+a_{Y}^{2}+a_{Z}^{2}}=g.$$

Once the initial values of *Ψ*, *θ*, and *φ* are determined, the initial orientation quaternion, $q_{t=0}$ is constructed from Equations (13)–(16).

The rate gyro angular velocities ($\omega_{X},\;\omega_{Y},\;\omega_{Z}$) are transformed into the fixed FOR by Equation (6), using the orthogonal form of the rotation matrix of Equation (9)
(17)$$\;[\begin{array}{c} \omega_{x}\\ \omega_{y}\\ \omega_{z} \end{array}]=\;[\begin{array}{ccc} q_{0}^{2}+q_{1}^{2}-q_{2}^{2}-q_{3}^{2} & 2\left(q_{1}q_{2}-q_{3}q_{0}\right) & 2\left(q_{1}q_{3}+q_{2}q_{0}\right)\\ 2\left(q_{1}q_{2}+q_{3}q_{0}\right) & q_{0}^{2}-q_{1}^{2}+q_{2}^{2}-q_{3}^{2} & 2\left(q_{2}q_{3}-q_{1}q_{0}\right)\\ 2\left(q_{1}q_{3}-q_{2}q_{0}\right) & 2\left(q_{2}q_{3}+q_{1}q_{0}\right) & q_{0}^{2}-q_{1}^{2}-q_{2}^{2}+q_{3}^{2} \end{array}]\;[\begin{array}{c} \omega_{X}\\ \omega_{Y}\\ \omega_{Z} \end{array}].$$

The incremental rotation quaternion, $p_{t}$ represented by these angular velocities at time *t* for a duration ∆*t*, is constructed using Equation (5)
(18)$$p_{t}=\cos\frac{\alpha }{2}+\sin\frac{\alpha }{2}\left(\omega_{x}i+\omega_{y}j+\omega_{z}k\right)/\|\omega \|$$
where
$$\|\omega \|=\sqrt{\omega_{x}^{2}+\omega_{y}^{2}+\omega_{z}^{2}}$$
and the rotation angle, *α*, is given by
$$\alpha =\Delta t\sqrt{\omega_{x}^{2}+\omega_{y}^{2}+\omega_{z}^{2}}.$$

From Equation (4) it then follows that
(19)$$q_{t+\Delta t}=p_{t}q_{t}.$$

We now have the orientation of the object as a function of time as the quaternions, $q_{t}$, and can use them to transform the acceleration vectors into the fixed FOR by the same process as was used for the angular velocities: (20)$$\left[\begin{array}{c} a_{x}\\ a_{y}\\ a_{z} \end{array}\right]=\;[\begin{array}{ccc} q_{0}^{2}+q_{1}^{2}-q_{2}^{2}-q_{3}^{2} & 2\left(q_{1}q_{2}-q_{3}q_{0}\right) & 2\left(q_{1}q_{3}+q_{2}q_{0}\right)\\ 2\left(q_{1}q_{2}+q_{3}q_{0}\right) & q_{0}^{2}-q_{1}^{2}+q_{2}^{2}-q_{3}^{2} & 2\left(q_{2}q_{3}-q_{1}q_{0}\right)\\ 2\left(q_{1}q_{3}-q_{2}q_{0}\right) & 2\left(q_{2}q_{3}+q_{1}q_{0}\right) & q_{0}^{2}-q_{1}^{2}-q_{2}^{2}+q_{3}^{2} \end{array}]\;[\begin{array}{c} a_{X}\\ a_{Y}\\ a_{Z} \end{array}].$$

These may then be numerically integrated with respect to time to get velocity and displacement in Cartesian space:(21)$$\dot{x}=\int_{0}^{t}a_{x}dt,\;x=\int_{0}^{t}\dot{x}dt$$
(22)$$\dot{y}=\int_{0}^{t}a_{y}dt,\;y=\int_{0}^{t}\dot{y}dt$$
(23)$$\dot{z}=\int_{0}^{t}a_{z}dt,\;z=\int_{0}^{t}\dot{z}dt$$

That completes the description of the object position, velocity, and orientation as a function of time.

### Other Practical Considerations

When these sensor packages are employed to track explosive- or shock-driven objects, the dynamic range of the data is often very large. As a result, the sensitivity of the sensor elements, especially the accelerometers, is sub-optimal at small accelerations, and the bias drift, which seems to scale with dynamic range, is significant at small accelerations. Even when the data are zeroed before the event, the bias changes as a result of the large accelerations during the event, and the measured values often do not return to zero. This results in non-zero post-event velocities and non-stationary end states, requiring some user intervention in interpreting the data. The rate gyros seem to be more robust, and generally return to zero.

## 3. An Illustrative Example

The motion of a large (≈1.4 m long), heavy conical test object subjected to an extreme air blast impulse (2 psi s, 13.8 kPa s) then needed to be measured. The impulse was generated from the detonation of a large mass of explosive in a 46-m-long, 2.4-m-diameter shock tube. The object was suspended at the open end of the shock tube from thin steel ropes that were severed 10 ms before the detonation was triggered. In this way, the object was in free fall at the time the blast wave arrived. The object has an initial 10.5 degree nose down pitch so that the impulse accelerated it both backwards and downwards into a soft-catch pit. Late-time luminescent detonation products and dust picked up by the blast prevented the optical tracking of the test object for an extended duration.

Instead, a diversified technical systems (DTS) slice nano (www.dtsweb.com (accessed on 11 November 2021)) and 6-axis sensor (part number Pro 2K-300) was bolted to a rigid mounting point inside the test object at a known location. This autonomous slice package and associated battery is capable of being triggered from either an externally supplied electrical signal or an internally generated trigger from detection of a rapid change in acceleration or angular attitude. All data are stored on the slice unit in non-volatile memory for later recovery. For this test, a primary electrical trigger coincident with the detonation was used with a software generated signal enabled as a backup in case of trigger signal failure.

The 6-axis sensor includes three orthogonal accelerometers with a range of ±2000 g and three angular rate gyros on the same axis orientation with a range of ±300 degrees/s. Data were recorded at 100 kHz and low-pass filtered using the recommended filter in the DTS control software prior to export as comma-separated with ASCII text files. Analysis and animation of the results were undertaken using the Igor software suite (www.wavemetrics.com (accessed on 11 November 2021)) although other common packages, such as Matlab or Python, could also be used.

As explained previously, the rate gyro data generally return to the initial zero bias position after blast loading. Therefore, these data are successfully processed with no user intervention. In this case, the relatively low range accelerometers used in the 6-axis module returned to within 0.25 g of their initial bias position. Therefore, minimal data correction was required to force the object to be at rest at a time when all real motion had ceased.

We discovered that the larger range acceleration modules used on other even more violent tests (e.g., ±20,000 g) often shift bias by up to 3 g after a test and significant effort is required to modify the acceleration data in a physically justifiable manner so that, for motions of several seconds, the object appears to come to rest. This is a result of the requirement that the total time integral of acceleration in each axis at the end of the test is zero to prevent apparent, but non-physical, residual velocities.

The simplest correction utilizes the following assumptions: (A) that a small non-zero offset is outputted from the accelerometer amplifier prior to the loading event which is safely subtracted from all data points prior to any analysis; and (B) that a single large and rapid loading instantly shifts the baseline to a different value for each axis which remains constant after the object comes to rest. The measured residual offset, corrected for the final orientation relative to the earth’s gravity, is then subtracted from all accelerometer data acquired after the first rapid motion is detected. This works quite well for single and brief rapid loading scenarios with relatively small subsequent values of acceleration (e.g., a single blast loading followed by a soft catch). It will be inadequate when a number of acceleration events occur in rapid succession since each event apparently produces a new temporary baseline shift until the next event. Without the object being at rest between acceleration events, these temporary offsets cannot be determined nor, therefore, subtracted.

The raw angular rate data in the sensor’s FOR for the blast loaded object are shown in Figure 2. The brief positive saturation at approximately 0.35 s was not found to affect the measured final resting orientation compared with the calculated one by a measurable amount (<3°). Figure 3 shows the roll, pitch, and yaw of the object over the motion time of approximately 2.05 s after applying the quaternion data reduction scheme described above. Although it is clear from inspection that the object yawed almost 180 degrees during the test before coming to rest, and from the pitch curve, that it gently bounced several times, it is difficult to intuitively translate this type of data into a physical interpretation of the motion. Of course, the raw data displayed in Figure 2 are even more difficult to interpret.

Instead, it is useful to animate the motion of a test object, and Figure 4 shows five example frames that correspond with the data presented in Figure 3. Initially, the object can be seen with a 10.5° nose down pitch and the action of the blast throws the object with only a modest decrease in pitch until it interacts with the soft-catch pit at about 0.25 s. From this time, it bounces and spins in a more complex fashion before coming to rest at the end of the pit with a positive pitch attitude. The full animation is shown in the Appendix A

## 4. Conclusions

We present a practical way of analyzing 6-axis accelerometer and rate gyro data from dynamic experiments, such as blast loading events where it is desirable to track an object’s motion in a challenging environment. Owing to the short duration of these types of tests, the corrections normally applied in applications, such as vehicle motion, to correct for long-term drift in these types of sensors are not applicable. However, we discovered that the rate gyro data generally return close enough to their initial zero bias measurement that these data may be processed without any user intervention and the output is accurate. In most cases, the data from high-quality, moderate-range accelerometers (≤2000 g) can be corrected for drift relatively easily. Data from longer duration tests (seconds) requiring the use of high-range accelerometers can be challenging to correct, although it is often the case that the still-accurate data at early times when minimal displacement has occurred is often of most practical relevance and so the technique still proves to be useful. 

## Figures and Tables

**Figure 1 sensors-21-07658-f001:**
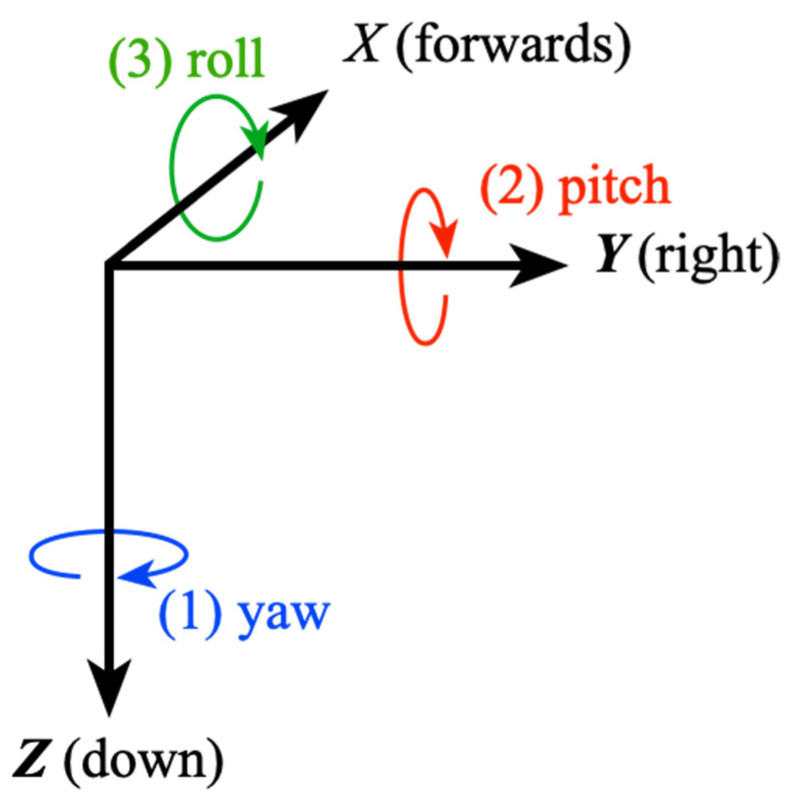
Tait–Bryan intrinsic rotations in the object’s FOR.

**Figure 2 sensors-21-07658-f002:**
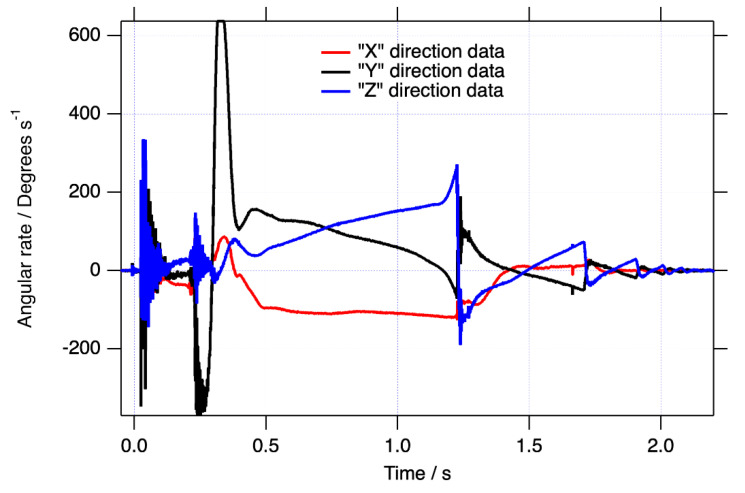
The raw angular rate data in the sensor’s FOR.

**Figure 3 sensors-21-07658-f003:**
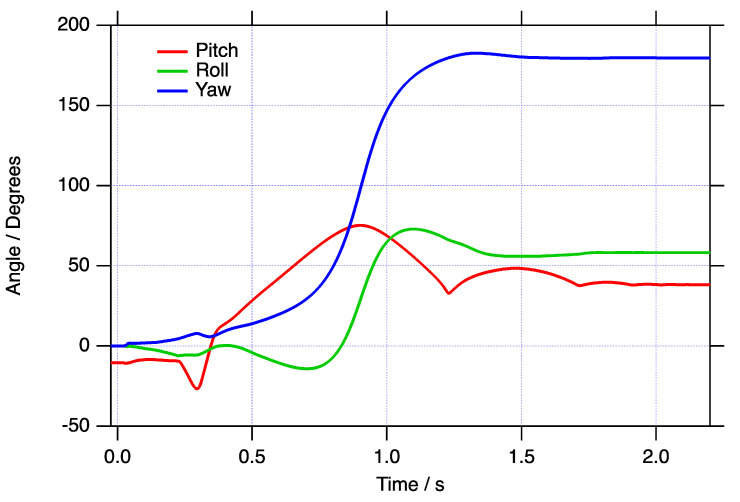
The roll, pitch, and yaw of the blast loaded object as a function of time. The object had an initial nose down pitch of 10.5° in the earth’s FOR.

**Figure 4 sensors-21-07658-f004:**
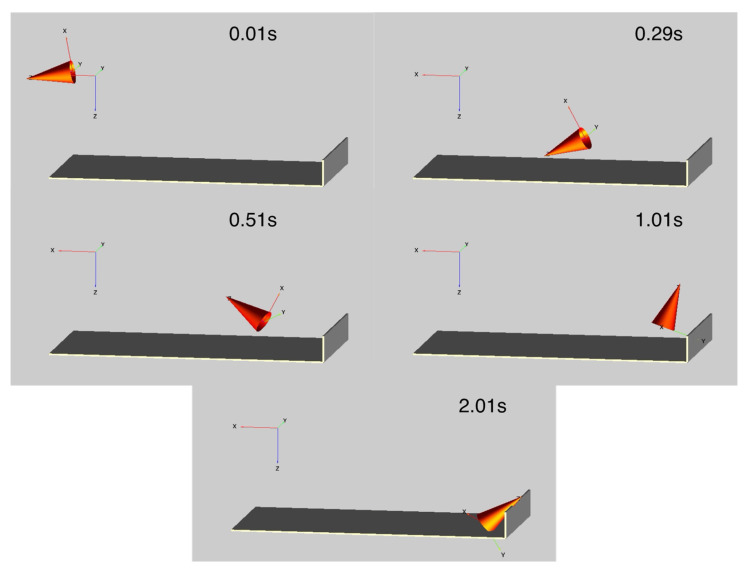
Selected animation frames showing the motion of the test object being thrown into the soft-catch pit. Time of 0.01 s shows the initial location prior to movement and 2.01 s shows the final resting location.

## Data Availability

The data presented in this study are available on request from the corresponding author. The data are not publicly available due to software licensing concerns.

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
