# Peer review of "The Application of Quaternions to Strap-Down MEMS Sensor Data"

_sensors, 2021, doi:10.3390/s21227658_

Round 1

Reviewer 1 Report

There was a mix of 3rd person passive and 1st person active that put the tone of the paper at odds with itself. Active is easier to read, but the authors should choose a single voice.

Examples:

Line 201: It was required to measure...

Line 229: We have discovered...

Of course, Line 229 could more simply read "We discovered..." to make active voice.

It seems a single experiment is detailed, and though the mathematics would remain the same for other experiments, the authors make statements about the adjustments that would be needed for low range accelerometers with respect to their initial bias position (Lines 226-227), but it is not clear from the text that this would be the extent of all such experiments. Nor does the paper address how this bias is accounted for in the mathematics governing the animation (e.g., is the bias only negated at the end or is it promulgated throughout the position/velocity calculations throughout the integrations?). Addressing this specifically would help.

Otherwise, the paper is very well written.

Author Response

There was a mix of 3rd person passive and 1st person active that put the tone of the paper at odds with itself. Active is easier to read, but the authors should choose a single voice.

Examples:

Line 201: It was required to measure...

Line 229: We have discovered...

Of course, Line 229 could more simply read "We discovered..." to make active voice.

We have addressed this issue and fixed a number of other minor style and typographical errors.

It seems a single experiment is detailed, and though the mathematics would remain the same for other experiments, the authors make statements about the adjustments that would be needed for low range accelerometers with respect to their initial bias position (Lines 226-227), but it is not clear from the text that this would be the extent of all such experiments. Nor does the paper address how this bias is accounted for in the mathematics governing the animation (e.g., is the bias only negated at the end or is it promulgated throughout the position/velocity calculations throughout the integrations?). Addressing this specifically would help.

Otherwise, the paper is very well written.

A short section discussing the easiest approach taken that has been found to be useful in some applications.

“The simplest correction utilizes the following assumptions: A) that a small non-zero offset is outputted from the accelerometer amplifier prior to the loading event and this is safely subtracted from all data points prior to any analysis and B) that a single large and rapid loading instantly shifts the baseline to a different value for each axis that remains constant after the object comes to rest. The measured residual offset, corrected for the final orientation relative to the earth’s gravity, is then subtracted from all accelerometer data acquired after the first rapid motion is detected. This works quite well for single and brief rapid loading scenarios with relatively small subsequent values of acceleration (e.g., a single blast loading followed by a soft catch). It will be inadequate when a number of acceleration events occur in rapid succession since each event apparently produces a new temporary baseline shift until the next event. Without the object being at rest between acceleration events these temporary offsets cannot be determined nor, therefore, subtracted.”

Reviewer 2 Report

A well written paper on use of quaternions for strap-down MEMS data. The concept is not novel, a number of books on inertial navigation provide a good explanation of the method. I used to have a class on this topic 2 decades ago. Software libraries are also readily available (Matlab, etc)

The application is interesting and unique though. This is where I would like to see more information being shared especially about the experimental setup. More experiments would definitely strengthen this work. 

The raw sensor data shall be presented along the global frame position changes in time (quaternion output).

The conclusion shall be expanded to reflect more the experimental findings.

Author Response

A well written paper on use of quaternions for strap-down MEMS data. The concept is not novel, a number of books on inertial navigation provide a good explanation of the method. I used to have a class on this topic 2 decades ago. Software libraries are also readily available (Matlab, etc)

The application is interesting and unique though. This is where I would like to see more information being shared especially about the experimental setup. More experiments would definitely strengthen this work. 

The raw sensor data shall be presented along the global frame position changes in time (quaternion output).

The conclusion shall be expanded to reflect more the experimental findings.

As explained in the comments to reviewer 3, every piece of the analysis has appeared multiple times in other references; however, we could not find a single reference that explained the steps required from raw data to useful output. We thought such a summary useful to others. Although not Matlab users, we generally find that unless extraordinarily well documented, reverse engineering mathematical steps from source code can be a time consuming business. Using the code as a ‘black box’ can be problematic.

Following the suggestion, we have added another figure showing the raw sensor data.

Reviewer 3 Report

First, the type of manuscript is Communication in email but Article in manuscript. Please determine the exact type of manuscript. Second, the introduction to quaternions is too much. Third, user intervention in interpreting the data mentioned in line 198 is not described in details. In general,the innovation is not enough. The manuscript focuses on the application of quaternions. However, the formulas are similar to the basic theory of quaternions. If possible, please describe the application of quaternion in combination with a set of specific data. Besides, is it necessary to compare the performance of other methods?

Author Response

First, the type of manuscript is Communication in email but Article in manuscript. Please determine the exact type of manuscript. Second, the introduction to quaternions is too much. Third, user intervention in interpreting the data mentioned in line 198 is not described in details. In general, the innovation is not enough. The manuscript focuses on the application of quaternions. However, the formulas are similar to the basic theory of quaternions. If possible, please describe the application of quaternion in combination with a set of specific data. Besides, is it necessary to compare the performance of other methods?

The confusion came about because it was originally submitted as an article, but it was suggested that it was really a communication. We have amended the first page to reflect the fact that it is a communication.

See the comments to reviewer 1 about the simple correction approach we have found useful in some circumstances.

We agree that the introduction and derivation section is not new; however, we spent some considerable time looking but could not find a logical step by step approach laid out in one report. Pieces are available in many sources but the interested researcher was forced to stitch them together themselves to successfully take the 6-axis raw data and turn it into a useful earth frame of reference. We think that this concatenation of steps will be useful to others.

The following has therefore been added to the introduction by way of explanation.

“Although all the mathematical derivations described in what follows are available in separate pieces in prior reports and books, we have not found an approachable and rigorous raw data input to earth FOR and thought such a summary would prove useful to others. Additionally, available descriptions for longer duration events often already include the necessary correction terms for the Kalman filter and this adds considerable but in this case unnecessary complexity to the equations.”

We have also added the following to the introduction to explain the paucity of other available techniques.

“The use of a 6-axis MEMS system safely contained within the test article when undertaking blast loading is a considerable advantage. The shock wave, explosive reaction products or dust kicked up by the blast are major problems for the only comparable technique for resolving full motion in 3-dimensions: orthogonal high-speed image analysis. Optical techniques in these conditions suffer from potential camera damage, severe image obstruction from smoke and dust and positioning suitable long duration high-intensity ancillary lighting. Additionally, motion resolution from image analysis is a complex tradeoff between frame exposure- and interframe-time and the dimensions of the field of view. Finally, the double differentiation required of measured object displacements to yield initial accelerations results in large uncertainties in the measured values during high magnitude and short duration events such as blast loading.”

Round 2

Reviewer 3 Report

I think the manuscript can be accepted after modifying.